# Annealing Effects on SnO_2_ Thin Film for H_2_ Gas Sensing

**DOI:** 10.3390/nano12183227

**Published:** 2022-09-16

**Authors:** Yijun Yang, Bohee Maeng, Dong Geon Jung, Junyeop Lee, Yeongsam Kim, JinBeom Kwon, Hee Kyung An, Daewoong Jung

**Affiliations:** 1Advanced Mechatronics R&D Group, Korea Institute of Industrial Technology (KITECH), Daegu 42994, Korea; 2School of Electronic and Electrical Engineering, College of IT Engineering, Kyungpook National University, 80 Daehakro, Daegu 41566, Korea

**Keywords:** SnO_2_ thin films, metal oxide gas sensor, hydrogen gas sensor, annealing effect

## Abstract

Hydrogen (H_2_) is attracting attention as a renewable energy source in various fields. However, H_2_ has a potential danger that it can easily cause a backfire or explosion owing to minor external factors. Therefore, H_2_ gas monitoring is significant, particularly near the lower explosive limit. Herein, tin dioxide (SnO_2_) thin films were annealed at different times. The as-obtained thin films were used as sensing materials for H_2_ gas. Here, the performance of the SnO_2_ thin film sensor was studied to understand the effect of annealing and operating temperature conditions of gas sensors to further improve their performance. The gas sensing properties exhibited by the 3-h annealed SnO_2_ thin film showed the highest response compared to the unannealed SnO_2_ thin film by approximately 1.5 times. The as-deposited SnO_2_ thin film showed a high response and fast response time to 5% H_2_ gas at 300 °C of 257.34% and 3 s, respectively.

## 1. Introduction

Since the industrial revolution, fossil fuels have been used as a major energy source for humanity. However, environmental problems caused by the emission of harmful gases during fuel combustion have emerged, and research on future renewable energy sources, such as solar, nuclear, and wind power, is underway. Among them, hydrogen (H_2_) is attracting attention as a renewable energy source used in various fields, such as automobiles, fuel cells, and chemical industries, because of its environmental friendliness and high energy density characteristics [1,2]. However, H_2_ has a flame propagation speed of approximately eight times faster than methane, whereas its minimum ignition energy is approximately 0.06 times lower than methane, which can easily cause a backfire or explosion owing to minor external factors [1,3,4]. To use the H_2_ gas in the industry environment, H_2_ stability should be ensured, and the detection of gases in the environment containing H_2_ is inevitable. H_2_ is a colorless, odorless, and volatile gas with a relatively high lower explosive limit of 4%. Therefore, detection by concentration at low and high concentrations and fast response time are significant in H_2_ sensing [3].

Based on the method of detection, H_2_ gas sensors are generally divided into contact (electrochemical, semiconductor, and solid electrolyte sensor), composite (photoionization and total reflection sensor), and optical type sensors (non-dispersive infrared, and photoacoustic sensor) [1,5]. The semiconductor gas sensor detects the target gas by reading a change in the resistance owing to the difference in the electronic density of a semiconductor surface of the sensing unit when gas flows. This sensing method is advantageous because semiconductor gas sensors are inexpensive, can be easily manufactured in large quantities through microelectromechanical system processes, has a simple configuration of detection circuits, and can detect various types of gases. However, because semiconductor gas sensors are driven at relatively high temperatures, they mainly use a metal-oxide-semiconductor form that is stable at high temperatures. Materials commonly used for metal oxide gas sensors include ZnO [4], SnO_2_ [6,7,8], and TiO_2_ [1], among which SnO_2_ has been extensively studied as SnO_2_ film-structured gas sensors exhibit high response results and fast response rates for specific gases, owing to their unique characteristics. SnO_2_ is an n-type semiconductor with a wide energy band gap of 3.6 eV and a tetragonal rutile structure caused by oxygen vacancy. In an n-type semiconductor such as SnO_2_, when external thermal energy is applied, an oxygen vacancy layer acts as a donor to change the number of electrons, thereby adjusting the grain boundary that influences the electrical conductivity and gas sensitivity of the semiconductor film. Therefore, SnO_2_ is mainly studied to adjust the electrical conductivity and gas sensitivity by controlling the thickness and grain size of the SnO_2_ thin film. In 1971, Naoyshi studied the first gas sensor that used SnO_2_ as a sensing material [6]. Thenceforth, various studies have been reported for SnO_2_ deposition methods, such as the sol-gel method [6], thermal evaporation [9], radio frequency (RF) sputtering [10], chemical vapor deposition [11], and the ion-beam method [12]. Among them, RF sputtering has been extensively investigated with the commercial advantages of a uniform deposition rate, cost-effectiveness, and high reproducibility. However, high energy deposition can result in unevenness or damage to the film. To improve performance and complement disadvantages of SnO_2_ film by sputtering, various post-treatment methods, such as annealing [13,14], acid treatment [15], and plasma treatment [16], were studied. Among them, the annealing method influences the shape and size of the crystal grains of the SnO_2_ thin film, changes the oxygen vacancy by the injected O_2_ gas in the annealing process, and changes the characteristics of the thin film.

In this study, the SnO_2_ thin film was deposited on a glass substrate using RF sputtering and applied as an H_2_ gas sensor. Our research aims to develop a high-performance sensor that can detect H_2_ more sensitively than general SnO_2_ sensors by optimizing the annealing time condition of SnO_2_ thin films and exploring the sensor operating temperature. In addition, based on the optimization conditions, it was attempted to maximize the H_2_ sensing ability by minimizing the problem of interference with other gases, which most metal oxide-based MOS sensors face.

## 2. Materials and Methods

### 2.1. Sensor Fabrication

The SnO_2_ thin film was deposited using RF sputtering. A glass wafer was used as the substrate, and Au was deposited using E-beam evaporation in advance to form an interdigitated electrode. The 99.99% purity target of SnO_2_ was used as a sputtering source under a base pressure of 5 × 10^−6^ torr. Ar and O_2_ gases were used as reaction gases under a flow rate of 10 sccm, and deposition energy of 150 W was applied. The SnO_2_ thin film, with dimensions of 2.7 × 2.85 mm, was deposited on the glass substrate, with a dimension of 5 × 5 mm. Subsequently, the deposited SnO_2_ thin film was annealed at 500 °C under air condition, which was annealed for 1~5 h, with intervals of 2 h. The schematic image of the full fabrication is shown in Figure 1.

### 2.2. Sensor Characterization

The X-ray diffraction (XRD, model Empyrean, Panalytical) was used to analyze the crystal structure before and after annealing of the deposited SnO_2_ thin film, and a field emission scanning electron microscope (model SU8220, Hitachi) was used for surface morphology and film thickness analysis. The atomic force microscope (AFM, model NX20, Park Systems) was used for surface characterization, and the hydrogen gas sensing performance was evaluated in the manufactured gas chamber, as shown in Figure 2. In the experiment, H_2_ gas concentration was adjusted using a mass flow controller (MFC), and a voltage of 1 V was applied to the gas sensor using a source meter (model B2902B, Keysight) to determine the changes in resistance during sensing caused by changes in the H_2_-air environment. The test was performed by fixing the air and H2 flow rate sum at 400 sccm using an MFC and adjusting the H2 concentration ratio. The gas detection experiment was performed by placing the sensor on the ceramic heater in the chamber and raising the temperature of the heater from 150 to 300 °C.

## 3. Results and Discussion

### 3.1. Structural and Morphological Characteristics

The crystalline structure is closely related to surface morphology. Based on the annealing process, gas sensing properties are influenced by inducing changes in morphology and structure. In addition, because the sensing process is usually operated at high temperatures, sensors that are annealed after deposition are more stable at the operating temperatures. Metal oxides such as SnO_2_ are highly sensitive to annealing, with different crystalline defects, such as oxygen vacancy or lattice disorder [17]. The grain size of the metal oxide shows a grown pattern under conditions of increasing the annealing time at a fixed temperature or changing the annealing temperature. In our case, the grain morphology change of SnO_2_ was confirmed by changing the annealing time under the constant temperature condition of 500 °C. The annealing temperature was set through experimental results. SnO_2_ thin films were prepared by the RF sputtering method, and these were annealed at a temperature of from 300 to 900 °C per 200 °C, respectively. The optimal temperature condition was set by measuring the response to the hydrogen of this thin film sensor. The finally set annealing optimum temperature was 500 °C. (Appendix A).

XRD patterns show the diffraction peaks of all the samples observed in the SnO_2_ pattern (JCPDS no. 41-1445, Figure 3a). From the XRD peaks, the average grain sizes of SnO_2_ thin films for different annealing times were calculated using the Scherrer equation:(1)$$D=\frac{Κ\lambda }{\beta \cos\theta }$$
where D is the grain size; Κ is a constant that depends on the crystalline shape and indices with a value of approximately 0.9; λ is the radiation wavelength; β is the crystalline size, measured at half of the maximum intensity; θ is the Bragg angle at which peaks are observed [18]. From the Scherrer equation, the average grain size of a SnO_2_ thin film was calculated and is denoted in Figure 3b.

As shown in Figure 3b, an increase in annealing time increased SnO_2_ grain size. Increases in annealing time result in corresponding increases in grain size. This is because smaller grains are combined into larger grains through high-temperature melting. The annealing time results in a rapid decrease in a non-bridging oxygen defects concentration, thereby forming SnO_2_ grains [19]. Therefore, an increase in annealing time can be a factor in increasing grain size [20,21,22].

The surface morphologies of SnO_2_ thin films from the as-deposited to 5-h annealed thin films, according to annealing time, are shown by the AFM images. As shown in Figure 4, the surface roughness decreased after the annealing time exceeded 1 h. The decrease was more severe when the annealing time exceeded 3 h. The thickness of film which is not annealed SnO_2_ was about 55.6 nm by SEM cross-section image (Appendix A). The surface roughness of the as-deposited SnO_2_ thin film was 1.042 nm. After annealing for 1 h, the surface roughness slightly increased to 1.066 nm. However, the surface roughness decreased as the annealing time increased, which was 0.921 nm after annealing the deposited SnO_2_ for 3 h. The surface roughness further decreased to 0.399 nm after annealing for 5 h, showing a certain difference compared to the others. After annealing for 1 h, the surface roughness increased because grains grew in the z-direction to the substrate surface, which occurred preferentially. Notably, island coalescence was absent for 1 h. However, after 1 to 5 h, the vertical surface roughness decreased with an increase in grain size, which became lateral. After being subjected to annealing for 1 h, the surface protrusions melted and grew on the lateral side, meaning the grain size increased. This decrease in surface roughness when the annealing time exceeded 1 h resulted from the island coalescence that created a smaller vertical distance to the top of the island [23].

The gas sensor based on an n-type metal-oxide-semiconductor, such as SnO_2_. is significantly influenced by the grain size. As the electrons flow against the energy barriers formed at the grain boundaries, the number of energy barriers decreases as the grain grows. When a SnO_2_ film deposited by sputtering was annealed, the SnO_2_ grain crumpled. Thus, the surface of the dense film became porous, increasing the reactive area for H_2_ gas that is accompanied by increased sensitivity of the SnO_2_ film. After annealing, the grain size of SnO_2_ increased, and the energy barriers and grain boundary defects decreased [24].

### 3.2. Sensing Performance

We validated the effect of SnO_2_ thin films on H_2_ by adjusting the annealing time and confirmed the optimal conditions in which the sensor will perform under various operating temperatures.

An experiment to evaluate the SnO_2_ sensing performance of H_2_ gas was conducted inside the gas chamber by adjusting the internal temperature and gas concentration. The sum of the total flow rates of H_2_ gas and air was fixed at 400 sccm, and the concentration of H_2_ gas was controlled using the MFC. In the experiment, the sensor response for H_2_ gas was calculated as follows:(2)$$\mathrm{Response},\;S\left(\%\right)=\frac{\Delta R}{R}\times 100=\frac{R_{0}-R}{R}\times 100,$$

$R_{0}$ is the resistance of the sensor measured in the air atmosphere before the H_2_ gas injection, and R is the resistance of the sensor exposed to H_2_ gas. Because SnO_2_ is an n-type semiconductor, the resistance at the time of reaction with the H_2_ gas is reduced. Therefore, the response of SnO_2_ for H_2_ gas is calculated as the difference in the resistance before the reaction to the change in the resistance after exposure to H_2_.

The sensitivity and response time of the 3 h and unannealed SnO_2_ for H_2_ gas as conducted in the gas chamber of a fixed internal temperature of 300 °C is shown in Figure 5. According to Figure 5a, the response of both 3-h and unannealed SnO_2_ increased with an increase in H_2_ concentration; however, the response of the 3-h annealed SnO_2_ showed a higher value than the unannealed SnO_2_ for the same hydrogen concentration. The gap between the values of the response of the two conditions increased as the difference in H_2_ concentration increased. When the H_2_ concentration reached 50,000 ppm, the highest response value was observed in the annealed condition (~378%), which was approximately 1.5 times higher than that of the unannealed (~251%). The H_2_ sensing response and recovery time of a 3-h annealed SnO_2_ as air and hydrogen flow is shown in Figure 5b. The response and recovery time of the 3-h annealed SnO_2_ were approximately 12 and 53 s, respectively, under H_2_ gas of 50,000 ppm at a chamber of 300 °C. (The graph of the response and recovery time of the sensor without annealing can be confirmed in the Appendix A). Kang et al. verified that the initial resistance of SnO2 increases at some point as the annealing temperature increases [25]. In here, we also confirmed that the resistance value in-creased from the initial annealing value of 35 K to 80 M during the performed annealing time of 1 to 5 hours (Appendix A). The response time was calculated as the elapsed time for the resistance difference between R_0_ and R to decrease from 90 to 10% after H_2_ gas injection. Similarly, the recovery time was calculated as the time elapsed for the resistance difference to change from 10 to 90%.

The sensitivity of metal oxides is influenced by many factors, such as the grain size of the material, surface area distribution, and charge carrier on the sensing surface. As shown in Figure 5, the sensitivity gradually increased with the annealing time, and the highest sensitivity was obtained at the 3-h condition. The response time showed a similar tendency to the sensitivity, and the highest value was obtained in the 3-h condition.

The sensitivity is closely related to the surface area and adsorption regions of the sensing gas. With an increase in grain size, the sensitivity of SnO_2_ also increases with the surface area. Therefore, the 3-h annealed SnO_2_ had a larger surface area and a larger grain size than the unannealed and 1-h-annealed SnO_2_ films. This increased the sensitivity (according to the previous measurements, the grain size of SnO_2_ increased with the annealing time (Section 3.1)). Figure 6 describes the charge transfer situation when the grain size is increased by performing annealing. In general, the electrical conductivity of semiconducting thin films is increased with annealing because, during the annealing treatment, it is influenced by the tunneling of the charge carriers through the barriers of the grain boundary and the recrystallization of grains [22,26]. The grain size is one of the significant factors that control resistance. The size of the crystallites is determined by directly manipulating the active surfaces (spread-out) of the thin films [22]. An increase in the diameter of the crystallites decreases the spread-out surface of the layer that is formed by the nanocrystallites’ decrease. The effect of annealing increases the extended (active) surface at a certain cleavage limit of the resulting crystal, and thus the resistance value decreases [22,27].

The response and recovery times were also expected to increase with the increasing grain size of SnO_2_ because a longer time is required to respond to more adsorption/desorption of H_2_ gas on the surface of the sensing materials. Moreover, a larger time is required for saturation and recovery to the initial values, as shown in Figure 7. Lee et al. validated that the heat treatment was essential to induce changes in the gas sensing film morphology that was more stable to be operated at the sensing temperature, and the response time increased when annealing was applied [19].

However, according to Figure 7, the response of the 5-h annealed SnO_2_ showed the lowest sensitivity, although the grain size increased. This lowered reactivity is considered to be a phenomenon caused by surface roughness, as shown in Figure 4. The total surface area of SnO_2_ annealed for 5 h was reduced compared to others, and the sensitivity to H_2_ was also reduced.

Metal oxide-based gas sensors have different performance values according to changes in the operating temperature, owing to their sensing principle. Moreover, the response time is highly influenced by the operating temperature. As shown in Figure 8a, the sensitivities of all the sensors, i.e., the 1, 3, and 5-h-annealed sensors, increased up to the operating temperature of 300 °C. Furthermore, the response time of the 3-h-annealed SnO_2_ decreased as the temperature increased, as shown in Figure 8b. Although the sensitivity and response time improved with an increase in operating temperature, the sensing tool was adversely influenced by the high operating temperature of the gas chamber. Therefore, the experiment was limited to temperatures of at most 300 °C.

Furthermore, we believe that the different optimal working temperatures for the sensors can be explained by assuming that two electron barriers exist between the SnO_2_ grains and Au electrodes [28]. One is the barrier between individual SnO_2_ grains, and the other is between the SnO_2_ and Au electrodes. A charge transfer by gas adsorption/desorption must traverse the two barriers to the measuring device (multimeter). Thus, as the temperature increases, the kinetic energy of the electrons increases, and more electrons can easily jump over the two barriers and move faster than at a lower temperature. Therefore, sensitivity and response time can be improved, as shown in Figure 8.

The selectivity of the SnO_2_ thin film to other gases was also validated. In addition to H_2_, CO_2_, NH_3_, and NO_2_ were reacted with the SnO_2_ thin sensor at 300 °C. The gas sensing mechanism of the metal oxide-based hydrogen detection sensor consists of a chemical reaction between chemically absorbed oxygen and the target gas. The selectivity results for each gas of the SnO_2_ thin film are shown in Figure 9. The response to CO_2_ with the same concentration of 50 ppm as H_2_ was the lowest at 3.89%. In the case of NO_2_ and NH_3_ gas, the sensor reacted with a low gas concentration of 50 and 20 ppm, respectively, owing to the risk of toxicity. The responses with these gases were 10.99% and 13.76%, respectively. Our gas sensor shows a high response to hydrogen. It is because our hydrogen sensor sets the optimal conditions for the high response, therefore exhibiting excellent reactivity compared to other gases. Another reason is that resistance deviation and the optimized temperature vary depending on the type of sensing gas. For example, NH_3_ shows the highest response for SnO_2_ thin film at about 100~150 °C. This temperature is much lower than the sensor operating temperature we set for the hydrogen detecting. This selectivity trend has shown that the optimized SnO_2_ sensor indicates a better response to hydrogen than NO_2_ and NH_3_ [28].

In sensor applications, results should be consistent for repeated gas reactions to establish the reliability of the measurement values and product commercialization. Repeatability tests for H_2_ gas were performed on a SnO_2_ thin film sensor annealed for 3 h. The result is shown in Figure 10. The tests were performed at a reaction temperature of 300 °C for 5% H_2_ gas, and a consistent gas response was shown without significant damage to the sensor, even after repeated reactions. A stable and reversible reaction was observed, even in the course of the repeated experiments. Based on these results, the SnO_2_ thin film sensor fabricated with the RF sputtering method and treated to an optimal annealing process was stable and suitable for use as a H_2_ detection sensor.

### 3.3. Sensing Mechanism

Gas sensing of SnO_2_ functions on the difference in resistance resulting from the properties of n-type semiconductors caused by free electrons inside. The conductivity of the semiconductor gas sensor is controlled by subsequent reactions, such as chemical adsorption, oxidation, and diffusion of oxygen on the tin oxide’s surface. This change occurs by gaining electrons from the reaction between absorbed oxygen and H_2_ [24].

If free electrons are activated by applying thermal energy to SnO_2_, they are captured on the surface by atmospheric oxygen, as shown in Figure 11a. By capturing the free electrons, the electron depletion layer is formed on the SnO_2_ surface, thereby increasing the potential barrier and the internal resistance, as shown in Figure 11b. The chemical equations that explain the capturing of free electrons by oxygen are as follows:(3)$$O_{2}\left(\mathrm{gas}\right)\to O_{2}\left(\mathrm{ads}\right),$$
(4)$$O_{2}\left(\mathrm{ads}\right)+e^{-}\to O_{2}^{-},$$
(5)$$O_{2}^{-}\left(\mathrm{ads}\right)+e^{-}\to 2O^{-},$$

Subsequently, H_2_ gas is injected, free electrons react with captured oxygen to form water vapor, the oxygen adsorbed to the surface is removed, the captured free electrons are released, as shown in Figure 11c, potential barriers are lowered, and the internal resistance is reduced. A H_2_ sensing reaction is expressed as follows:(6)$$2H_{2}+O_{2}^{-}\left(\mathrm{ads}\right)\to 2H_{2}O+e^{-},$$
(7)$$2H_{2}+O^{-}\left(\mathrm{ads}\right)\to 2H_{2}O+e^{-},$$
(8)$$4H+O_{2}^{-}\left(\mathrm{ads}\right)\to 2H_{2}O+e^{-},$$

## 4. Conclusions

This study fabricated a SnO_2_ thin film gas sensor by depositing a metal oxide-based layer using the RF sputtering technique and annealing under high-temperature conditions. Here, the hydrogen gas detection performance of this SnO_2_ thin film sensor, according to the annealing time and gas reaction operating temperature, was demonstrated.

We experimentally validated the H_2_ sensing capability of the SnO_2_ thin film sensor to improve performance by increasing the metal oxide grain size as the annealing treatment time increased (up to 3 h), which led to the improvement of the surface area for the gas response. In addition, we validated that the improvement of the operating temperature of the gas reaction of the sensor can improve sensor performance and shorten the response time by assisting the electrons to move through the electronic barrier. Our sensor showed excellent selectivity in hydrogen gas compared to other gases. In addition, the sensor could operate without being damaged, even for repeated gas reactions.

This study demonstrated that a robust and stable metal oxide sensor could be manufactured by completing the optimal annealing treatment on the metal oxide thin film introduced by the RF sputtering technique.

## Figures and Tables

**Figure 1 nanomaterials-12-03227-f001:**
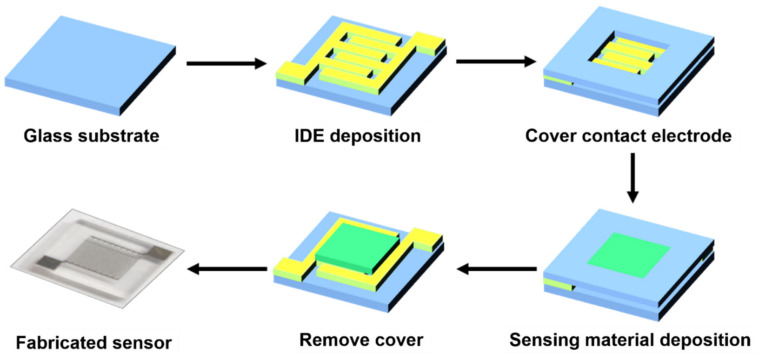
Schematic for SnO_2_ gas sensor fabrication.

**Figure 2 nanomaterials-12-03227-f002:**
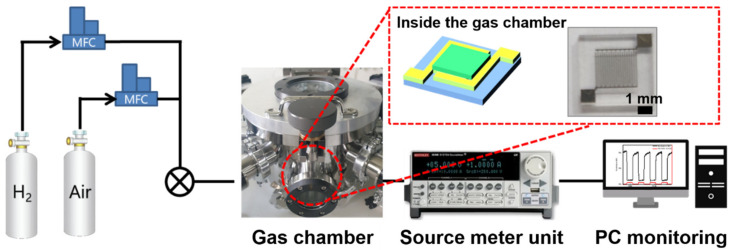
Schematic of gas sensing setup.

**Figure 3 nanomaterials-12-03227-f003:**
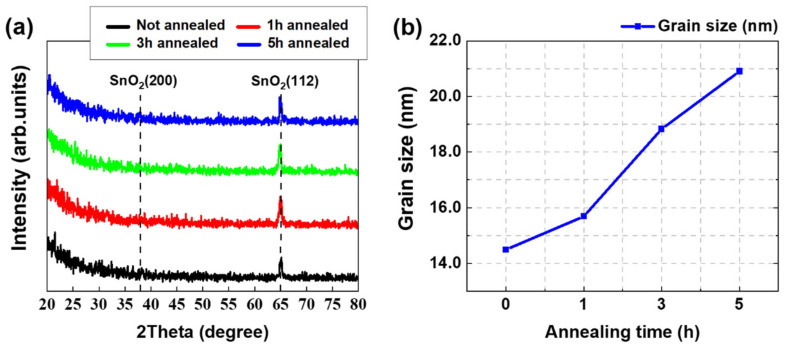
(**a**) XRD of SnO_2_ for annealing time; (**b**) average grain sizes of the SnO_2_ thin film with annealing times.

**Figure 4 nanomaterials-12-03227-f004:**
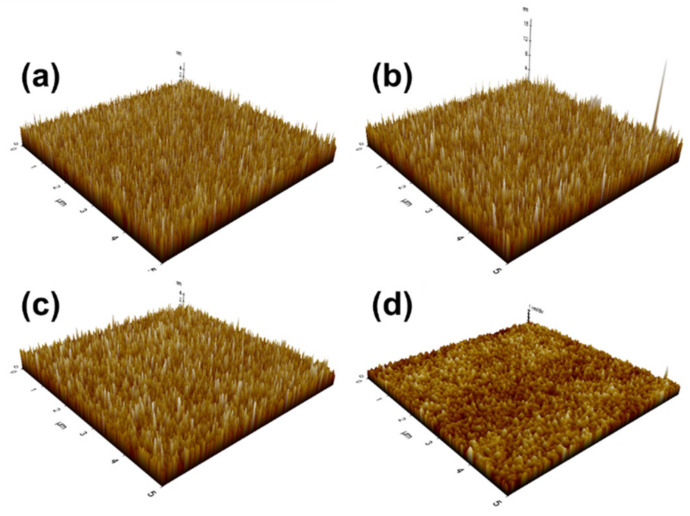
AFM surface image of SnO_2_ thin film by annealing time; (**a**) as-deposited not annealed; (**b**) 1 h. annealed (**c**) 3-h annealed (**d**) 5-h annealed.

**Figure 5 nanomaterials-12-03227-f005:**
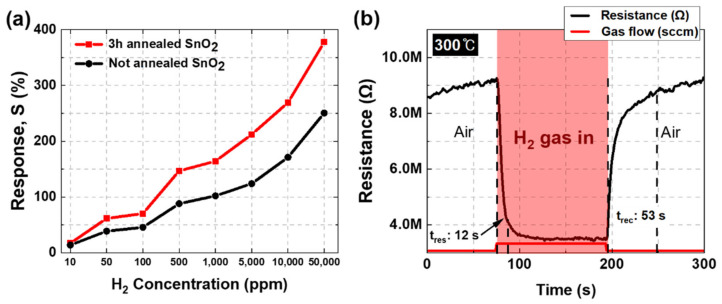
(**a**) Sensitivity of 3-h annealed SnO_2_ and not annealed SnO_2_ by H_2_ concentration; (**b**) hydrogen response and recovery time of 3-h annealed SnO_2_.

**Figure 6 nanomaterials-12-03227-f006:**
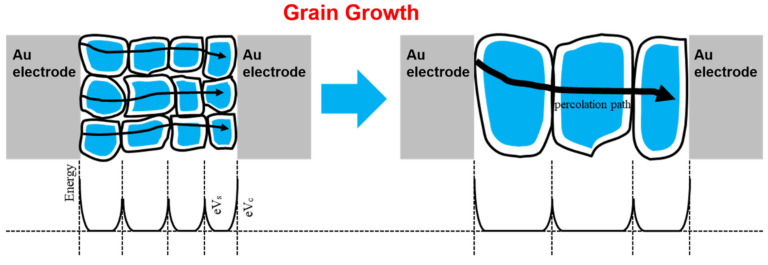
Schematic image of SnO_2_ grain boundary and Schottky barrier with Au electrode before and after annealing.

**Figure 7 nanomaterials-12-03227-f007:**
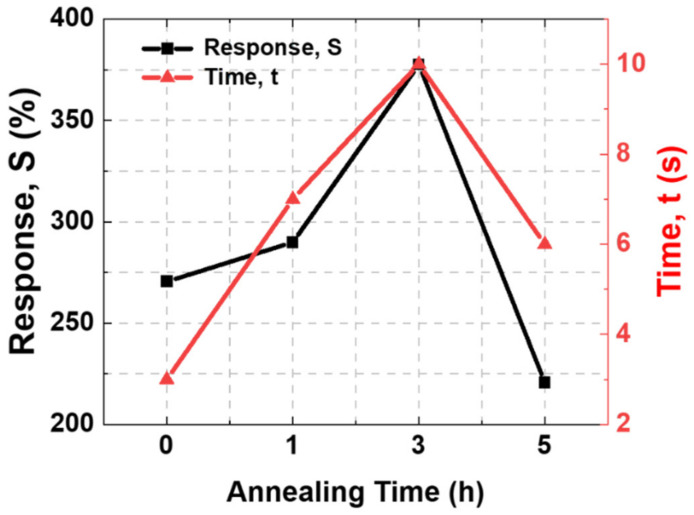
Sensitivity and response time for annealing time of SnO_2_ thin film under a 300 °C gas reaction condition.

**Figure 8 nanomaterials-12-03227-f008:**
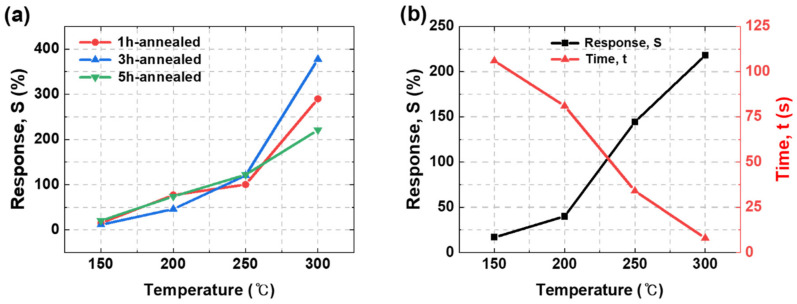
(**a**) Comparison of response according to the operating temperature of sensors fabricated under different annealing conditions. (**b**) Comparison of sensitivity and response time according to sensor operating temperature of H_2_ sensor fabricated by annealing for 3 h.

**Figure 9 nanomaterials-12-03227-f009:**
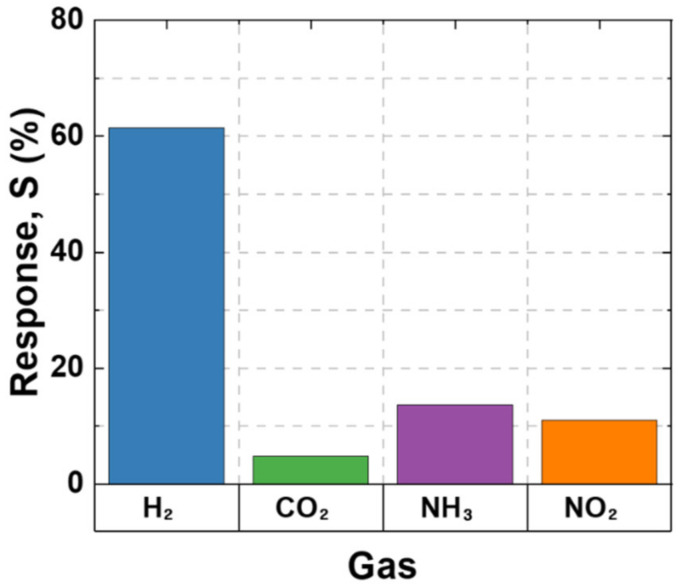
Gas selectivity of SnO_2_ thin film.

**Figure 10 nanomaterials-12-03227-f010:**
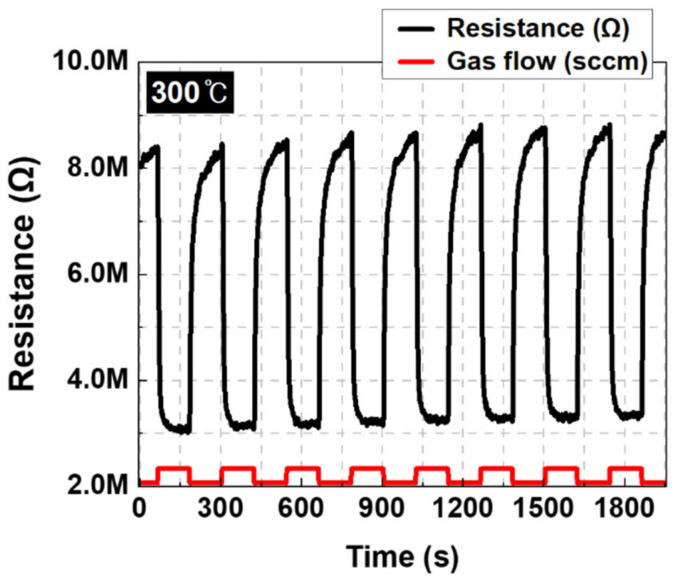
Repeatability tests of 3-h-annealed SnO_2_ thin film (5%).

**Figure 11 nanomaterials-12-03227-f011:**
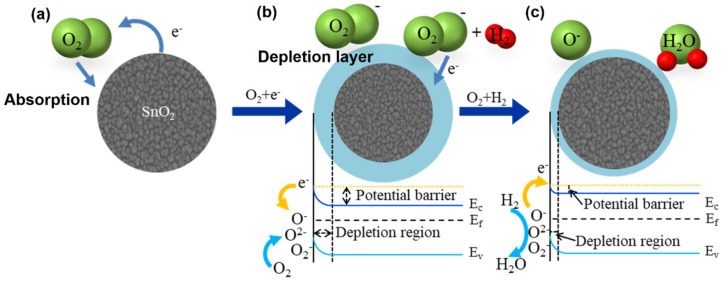
n—type metal oxide hydrogen sensing mechanism; (**a**) SnO_2_ as-deposited; (**b**) SnO_2_ in air; (**c**) SnO_2_ in H_2_ reduction gas.

## Data Availability

Not applicable.

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
