# Peer review of "Annealing Effects on SnO2 Thin Film for H2 Gas Sensing"

_nanomaterials, 2022, doi:10.3390/nano12183227_

Round 1
Reviewer 1 Report
The manuscript written by Y. Yang et al. reports the annealing effects of tin oxide thin film for hydrogen sensing. However, some questions still need to be solved. Therefore, I suggest that the manuscript could be accepted after minor revisions.
1. The H2 sensing experiment described by the author was carried out in a 300°C chamber. Is the whole chamber 300°C or the device? Please describe the experimental process in detail.
2. What is the thickness of the film? It is better to give a cross-sectional SEM image.
3. What is the change trend of the resistance of the film after annealing? Please give the data and explain.
4. The response of hydrogen in Figure 9 is higher than that of other interfering gases. Please explain the reason briefly. In addition, the resistance of tin oxide exposed to the reducing gas and the oxidizing gas should be decreased and increased respectively.
5. Please carefully check some errors in the text. For example, the number 5 is omitted in line 134, and Fig. 1 does not indicate in the main text.
Reviewer 2 Report
Dear Editor!
The manuscript «Annealing effects on SnO2 thin film for H2 gas sensing» submitted for consideration by Nanomaterials reports about development of H2 sensor based on SnO2 nanocrystalline thin film. The article is well structured and contains new, original, interesting results. There is clarification comment to the authors:
In Figure 9 authors compare selectivity of the SnO2 thin film to different gases. Is there a physical explanation for the increased sensitivity of the film specifically to hydrogen compared to other gases? In the text of the manuscript, it would be useful to comment on this issue.
I recommend accepting for publication after minor revision.
Reviewer 3 Report
This study reports the fabrication of a SnO2 thin film for gas sensing, through depositing a metal oxide-based 279 layer using RF sputtering technique, and then annealled under high temperatures. Various techniques were used for characterization, such XRD, SEM, AFM and sensing evaluating techinques. Some results and findings may be interesting to the readership in this area, and thus this article may be accepted, however, subjecting to addressing the following major issues:
1. The originality using SnO2 for detecting H2 is not addressed clearly in this work;
2. The effects of annealing temperatures, such as 300,400, 500, 600oC, should be provided and compared;
3. Why did the authors choose CO2, NO2 rather than CO, NO gas species for comparison, and the latter have similar reducting property as H2?
4. The detecting mechanisms for sensing CO2, NO2 gas species were not well explained;
5. More TEM images for showing the size changes of SnO2 particles with annealing temperatures should be provided in the revison;
6. The English writing of context should be further polished.
, Here, the hydrogen gas detection performance of this SnO2 thin film sensor according to 281 the annealing time and gas reaction operating temperature was demonstrated.
Reviewer 4 Report
Hydrogen (H2) is attracting attention as a renewable energy source in various fields, so gas sensing technology is good way for the safe application of this novel energy. In this manuscript, tin dioxide (SnO2) thin films were utilized as sensing structure for the gas sensor. There some work should be improved before be accepted for published.
1, what is effect of anealling time on the SnO2 thin films, what is theory of the gas sensing,
2, the curve in the figure 6, why the response has the highest value when the annealing time is 3hr,
3, are there the experimental results to verify the process in Figure 8.
Round 2
Reviewer 3 Report
The revision is suitable for accepting by the Journal.